# DISTRIBUTIONAL SLICED-WASSERSTEIN AND APPLICATIONS TO GENERATIVE MODELING

**Khai Nguyen**
VinAI Research, Vietnam
`v.khainb@vinai.io`

**Nhat Ho**[*]
University of Texas, Austin
VinAI Research, Vietnam
`minhnhat@utexas.edu`

**Tung Pham**
VinAI Research, Vietnam
`v.tungph4@vinai.io`

**Hung Bui**
VinAI Research, Vietnam
`v.hungbh1@vinai.io`

## ABSTRACT

Sliced-Wasserstein distance (SW) and its variant, Max Sliced-Wasserstein distance (Max-SW), have been used widely in the recent years due to their fast computation and scalability even when the probability measures lie in a very high dimensional space. However, SW requires many unnecessary projection samples to approximate its value while Max-SW only uses the most important projection, which ignores the information of other useful directions. In order to account for these weaknesses, we propose a novel distance, named *Distributional Sliced-Wasserstein* distance (DSW), that finds an *optimal* distribution over projections that can balance between exploring distinctive projecting directions and the informativeness of projections themselves. We show that the DSW is a generalization of Max-SW, and it can be computed efficiently by searching for the optimal push-forward measure over a set of probability measures over the unit sphere satisfying certain regularizing constraints that favor distinct directions. Finally, we conduct extensive experiments with large-scale datasets to demonstrate the favorable performances of the proposed distances over the previous sliced-based distances in generative modeling applications.

## 1 INTRODUCTION

Optimal transport (OT) is a classical problem in mathematics and operation research. Due to its appealing theoretical properties and flexibility in practical applications, it has recently become an important tool in the machine learning and statistics community; see for example, (Courty et al., 2017; Arjovsky et al., 2017; Tolstikhin et al., 2018; Gulrajani et al., 2017) and references therein. The main usage of OT is to provide a distance named Wasserstein distance, to measure the discrepancy between two probability distributions. However, that distance suffers from expensive computational complexity, which is the main obstacle to using OT in practical applications.

There have been two main approaches to overcome the high computational complexity problem: either approximate the value of OT or apply the OT adaptively to specific situations. The first approach was initiated by (Cuturi, 2013) using an entropic regularizer to speed up the computation of the OT (Sinkhorn, 1967; Knight, 2008). The entropic regularization approach has demonstrated its usefulness in several application domains (Courty et al., 2014; Genevay et al., 2018; Bunne et al., 2019). Along this direction, several works proposed efficient algorithms for solving the entropic OT (Altschuler et al., 2017; Lin et al., 2019b;a) as well as methods to stabilize these algorithms (Chizat et al., 2018; Peyré & Cuturi, 2019; Chizat et al., 2018; Schmitzer, 2019). However, these algorithms have complexities of the order $\mathcal{O}(k^2)$, where $k$ is the number of supports. It is expensive when we need to compute the OT repeatedly, especially in learning the data distribution.

---

[*]The work was finished when Nhat Ho worked at VinAI Research in the summer of 2020.

The second approach, known as "slicing", takes a rather different perspective. It leverages two key ideas: the OT closed-form expression for two distributions in one-dimensional space, and the transformation of a distribution into a set of projected one-dimensional distributions by the Radon transform (RT) (Helgason, 2010). The popular proposal along this direction is Sliced-Wasserstein (SW) distance (Bonneel et al., 2015), which samples the projecting directions uniformly over a unit sphere in the data ambient space and takes the expectation of the resulting one-dimensional OT distance. The SW distance hence requires a significantly lower computation cost than the original Wasserstein distance and is more scalable than the first approach. Due to its solid statistical guarantees and efficient computation, the SW distance has been successfully applied to a variety of practical tasks (Deshpande et al., 2018; Liutkus et al., 2019; Kolouri et al., 2018; Wu et al., 2019; Deshpande et al., 2019) where it has been shown to have comparative performances to other distances and divergences between probability distributions. However, there is an inevitable bottleneck of computing the SW distance. Specifically, the expectation with respect to the uniform distribution of projections in SW is intractable to compute; therefore, the Monte Carlo method is employed to approximate it. Nevertheless, drawing from a uniform distribution of directions in high-dimension can result in an overwhelming number of irrelevant directions, especially when the actual data lies in a low-dimensional manifold. Hence, SW typically needs to have a large number of samples to yield an accurate estimation of the discrepancy. Alternatively, in the other extreme, Max Sliced-Wasserstein (Max-SW) distance (Deshpande et al., 2019) uses only one important direction to distinguish the probability distributions. However, other potentially relevant directions are ignored in Max-SW. Therefore, Max-SW can miss some important differences between the two distributions in high dimension. We note that the linear projections in the Radon transform can be replaced by non-linear projections resulting in the generalized sliced-Wasserstein distance and its variants (Beylkin, 1984; Kolouri et al., 2019).

Apart from these main directions, there are also few proposals that try either to modify them or to combine the advantages of the above-mentioned approaches. In particular, Paty & Cuturi (2019) extended the idea of the max-sliced distance to the max-subspace distance by considering finding an optimal orthogonal subspace. However, this approach is computationally expensive, since it could not exploit the closed-form of the one-dimensional Wasserstein distance. Another approach named the Projected Wasserstein distance (PWD), which was proposed in (Rowland et al., 2019), uses sliced decomposition to find multiple one-dimension optimal transport maps. Then, it computes the average cost of those maps equally in the original dimension.

**Our contributions.** Our paper also follows the slicing approach. However, we address key friction in this general line of work: how to obtain a relatively small number of slices simultaneously to maintain the computational efficiency, but at the same time, cover the major differences between two high-dimensional distributions. We take a probabilistic view of slicing by using a probability measure on the unit sphere to represent how important each direction is. From this viewpoint, SW uses the uniform distribution while Max-SW searches for the best delta-Dirac distribution over the projections, both can be considered as special cases. In this paper, we propose to search for an optimal distribution of important directions. We regularize this distribution such that it prefers directions that are far away from one another, hence encouraging an efficient exploration of the space of directions. In the case of no regularization, our proposed method recovers max-(generalized) SW as a special case. In summary, our main contributions are two-fold:

1. First, we introduce a novel distance, named *Distributional Sliced-Wasserstein distance* (DSW), to account for the issues of previous sliced distances. Our main idea is to search for not just a single most important projection, but an *optimal* distribution over projections that could balance between an expansion of the area around important projections and the informativeness of projections themselves, i.e., how well they can distinguish the two target probability measures. We show that DSW is a proper metric in the probability space and possesses appealing statistical and computational properties as the previous sliced distances.

2. Second, we apply the DSW distance to generative modeling tasks based on the generative adversarial framework. The extensive experiments on real and large-scale datasets show that DSW distance significantly outperforms the SW and Max-SW distances under similar computational time on these tasks. Furthermore, the DSW distance helps model distribution converge to the data distribution faster and provides more realistic generated images than the SW and Max-SW distances.

**Organization.** The remainder of the paper is organized as follows. In Section 2, we provide backgrounds for Wasserstein distance and its slice-based versions. In Section 3, we propose distributional (generalized) sliced-Wasserstein distance and analyze some of its theoretical properties. Section 4 includes extensive experiment results followed by discussions in Section 5. Finally, we defer the proofs of key results and extra materials in the Appendices.

**Notation.** For any $\theta, \theta' \in \mathbb{R}^d$, $\cos(\theta, \theta') = \frac{\theta^\top \theta'}{\|\theta\| \|\theta'\|}$, where $\|.\|$ is $\ell_2$ norm. For any $d \geq 2$, $\mathbb{S}^{d-1}$ denotes the unit sphere in $d$ dimension in $\ell_2$ norm . Furthermore, $\delta$ denotes the Dirac delta function, and $\langle \cdot, \cdot \rangle$ is the Euclidean inner-product. For any $p \geq 1$, $\mathbb{L}^p(\mathbb{R}^d)$ is the set of real-valued functions on $\mathbb{R}^d$ with finite $p$-th moment.

## 2 BACKGROUND

In this section, we provide necessary backgrounds for the (generalized) Radon transform, the Wasserstein, and sliced-Wasserstein distances.

### 2.1 WASSERSTEIN DISTANCE

We start with a formal definition of Wasserstein distance. For any $p \geq 1$, we define $\mathcal{P}_p(\mathbb{R}^d)$ as the set of Borel probability measures with finite $p$-th moment defined on a given metric space $(\mathbb{R}^d, \|.\|)$. For any probability measures $\mu, \nu$ defined on $\mathcal{X}, \mathcal{Y} \subseteq \mathbb{R}^d$, we denote their corresponding probability density functions as $I_\mu$ and $I_\nu$. The Wasserstein distance of order $p$ between $\mu$ and $\nu$ is given by (Villani, 2008; Peyré & Cuturi, 2019):

$$W_p(\mu, \nu) := \Big( \inf_{\pi \in \Pi(\mu, \nu)} \int_{\mathcal{X} \times \mathcal{Y}} \|x - y\|^p d\pi(x, y) \Big)^{\frac{1}{p}},$$

where $\Pi(\mu, \nu)$ is a set of all transportation plans $\pi$ such that the marginal distributions of $\pi$ are $\mu$ and $\nu$, respectively. In order to simplify the presentation, we abuse the notation by using both $W_p(\mu, \nu)$ and $W_p(I_\mu, I_\nu)$ interchangeably for the Wasserstein distance between $\mu$ and $\nu$. When $\mu$ and $\nu$ are *one-dimension* measures, the Wasserstein distance between $\mu$ and $\nu$ has a closed-form expression $W_p(\mu, \nu) = (\int_0^1 |F_\mu^{-1}(z) - F_\nu^{-1}(z)|^p dz)^{1/p}$ where $F_\mu$ and $F_\nu$ are the cumulative distribution function (CDF) of $I_\mu$ and $I_\nu$, respectively.

### 2.2 (GENERALIZED) RADON TRANSFORMS

Now, we review (generalized) Radon transform maps, which are key to the notion of (generalized) sliced-Wasserstein distance and its variants. The *Radon transform* (RT) maps a function $I \in \mathbb{L}^1(\mathbb{R}^d)$ to the space of functions defined over space of lines in $\mathbb{R}^d$. In particular, for any $t \in \mathbb{R}$ and direction $\theta \in \mathbb{S}^{d-1}$, the RT is defined as follows (Helgason, 2010) : $\mathcal{R}I(t, \theta) := \int_{\mathbb{R}^d} I(x) \delta(t - \langle x, \theta \rangle) dx$.

The *generalized* Radon transform (GRT) (Beylkin, 1984) extends the original one from integration over hyperplanes of $\mathbb{R}^d$ to integration over hypersurfaces. In particular, it is defined as: $\mathcal{G}I(t, \theta) := \int_{\mathbb{R}^d} I(x) \delta(t - g(x, \theta)) dx$, where $t \in \mathbb{R}$ and $\theta \in \Omega_\theta$. Here, $\Omega_\theta$ is a compact subset of $\mathbb{R}^d$ and $g : \mathbb{R}^d \times \mathbb{S}^{d-1} \mapsto \mathbb{R}$ is a defining function (cf. Assumptions H1-H4 in (Kolouri et al., 2019) for the definition of defining function) inducing the hypersurfaces. When $g(x, \theta) = \langle x, \theta \rangle$ and $\Omega_\theta = \mathbb{S}^{d-1}$, the generalized Radon transform becomes the standard Radon transform.

### 2.3 (GENERALIZED) SLICED-WASSERSTEIN DISTANCES

The sliced-Wasserstein distance (SW) between two probability measures $\mu$ and $\nu$ is defined as (Bonneel et al., 2015): $SW_p(\mu, \nu) := (\int_{\mathbb{S}^{d-1}} W_p^p(\mathcal{R}I_\mu(\cdot, \theta), \mathcal{R}I_\nu(\cdot, \theta)) d\theta)^{1/p}$. Similarly, the generalized sliced-Wasserstein distance (Kolouri et al., 2019) (GSW) is given by $GSW_p(\mu, \nu) := (\int_{\Omega_\theta} W_p^p(\mathcal{G}I_\mu(\cdot, \theta), \mathcal{G}I_\nu(\cdot, \theta)) d\theta)^{1/p}$, where $\Omega_\theta$ is the compact set of feasible parameter. However, these integrals are usually intractable. Thus, they are often approximated by using Monte Carlo scheme to draw uniform samples $\{\theta_i\}_{i=1}^N$ from $\mathbb{S}^{d-1}$ and $\Omega_\theta$. In particular, $SW_p^p(\mu, \nu) \approx \frac{1}{N} \sum_{i=1}^N W_p^p(\mathcal{R}I_\mu(\cdot, \theta_i), \mathcal{R}I_\nu(\cdot, \theta_i))$ and $GSW_p^p(\mu, \nu) \approx \frac{1}{N} \sum_{i=1}^N W_p^p(\mathcal{G}I_\mu(\cdot, \theta_i), \mathcal{G}I_\nu(\cdot, \theta_i))$. In order to obtain a good approximation of (generalized) SW distances, $N$ needs to be sufficiently large.

However, important directions are not distributed uniformly over the sphere. Thus, this approach will draw potentially many unimportant projections that are not only expensive but also greatly reduce the effect of the SW distance.

### 2.4 MAX (GENERALIZED) SLICED-WASSERSTEIN DISTANCES

An approach to using only informative directions is to simply take the best slice in discriminating two given probability distributions. That distance is max sliced-Wasserstein distance (Max-SW) (Deshpande et al., 2019), which is given by $\max SW_p(\mu, \nu) := \max_{\theta \in \mathbb{S}^{d-1}} W_p(\mathcal{R}I_\mu(\cdot, \theta), \mathcal{R}I_\nu(\cdot, \theta))$. By combining this idea with non-linear projections from generalized Radon transform, we obtain max generalized sliced-Wasserstein distance (Max-GSW) (Kolouri et al., 2019). The formal definition of that distance is: $\max GSW_p(\mu, \nu) := \max_{\theta \in \Omega_\theta} W_p(\mathcal{G}I_\mu(\cdot, \theta), \mathcal{G}I_\nu(\cdot, \theta))$. The (generalized) Max-SW distances focus on finding only the most important direction. Meanwhile, other informative directions play no role in the distance. Therefore, (generalized) Max-SW distances can ignore useful information about the structure of high dimensional probability measures.

## 3 DISTRIBUTIONAL SLICED-WASSERSTEIN DISTANCE

With the aim of improving the limitations of the previous sliced distances, we propose a novel distance, named *Distributional Sliced-Wasserstein distance* (DSW), that can search for not just a single but a distribution of important directions on the unit sphere. We prove that it is a well-defined metric and discuss its connection to the existing sliced-based distances in Section 3.1. Then, we provide a procedure to approximate DSW based on its dual form in Section 3.2.

### 3.1 DEFINITION AND METRICITY

We first start with a definition of distributional sliced-Wasserstein distance. We say $C > 0$ *admissible* if the set $\mathbb{M}_C$ of probability measures $\sigma$ on $\mathbb{S}^{d-1}$ satisfying $\mathbb{E}_{\theta, \theta' \sim \sigma} \left[ |\theta^\top \theta'| \right] \leq C$ is not empty.

**Definition 1.** *Given two probability measures $\mu$ and $\nu$ on $\mathbb{R}^d$ with finite $p$-th moments where $p \geq 1$ and an admissible regularizing constant $C > 0$. The* distributional sliced-Wasserstein distance *(DSW) of order $p$ between $\mu$ and $\nu$ is given by:*

$$DSW_p(\mu, \nu; C) := \sup_{\sigma \in \mathbb{M}_C} \left( \mathbb{E}_{\theta \sim \sigma} \left[ W_p^p(\mathcal{R}I_\mu(\cdot, \theta), \mathcal{R}I_\nu(\cdot, \theta)) \right] \right)^{\frac{1}{p}}, \tag{1}$$

*where $\mathcal{R}$ is the Radon transform operator.*

The DSW aims to find the optimal probability measure of slices on the unit sphere $\mathbb{S}^{d-1}$. Note that, the Max-SW distance is equivalent to searching for the best Dirac measure on a single point in $\mathbb{S}^{d-1}$, which puts all weights in only one direction. Meanwhile, the uniform measure in the formulation of SW distance distributes the same weights in all directions. Indeed, the uniform and Dirac measures are two special cases, because they view that either all directions are equally important or only one direction is important. That view is too restricted if the data actually lie on low dimensional space. Thus, we aim to find a probability measure which concentrates only on areas around important directions. Furthermore, we do not want these directions to lie in only one small area, because under the orthogonal projection of RT, their corresponding one-dimensional distributions will become similar. In order to achieve this, we search for an optimal measure $\sigma$ that satisfies the regularization constraint $\mathbb{E}_{\theta, \theta' \sim \sigma}[|\theta^\top \theta'|] \leq C$. By Cauchy-Schwarz inequality, $C$ is no greater than 1, thus $\mathbb{M}_1$ contains all probability measures on the unit sphere. Optimizing over $\mathbb{M}_1$ simply returns the best Dirac measure corresponding to the Max-SW distance. When $C$ is small, the constraint forces the measure $\sigma$ to distribute more weights to directions that are far from each other (in terms of their angles). Thus, a small appropriate value of $C$ will help to balance between the distinctiveness and informativeness of these targeted directions. For further discussion about $C$, see Appendix B.1.

Next, we show that DSW is a well-defined metric on the probability space.

**Theorem 1.** *For any $p \geq 1$ and admissible $C > 0$, $DSW_p(\cdot, \cdot; C)$ is a well-defined metric in the space of Borel probability measures with finite $p$-th moment. In particular, it is non-negative, symmetric, identity, and satisfies the triangle inequality.*

The proof of Theorem 1 is in Appendix A.1. Our next result establishes the topological equivalence between DSW distance and (max)-sliced Wasserstein and Wasserstein distances.

**Theorem 2.** *For any $p \geq 1$ and admissible $C > 0$, the following holds*

    (a) $DSW_p(\mu, \nu; C) \leq maxSW_p(\mu, \nu) \leq W_p(\mu, \nu)$.

    (b) *If $C \geq 1/d$, we have* $DSW_p(\mu, \nu; C) \geq \left(\frac{1}{d}\right)^{1/p} maxSW_p(\mu, \nu) \geq \left(\frac{1}{d}\right)^{1/p} SW_p(\mu, \nu)$.

*As a consequence, when $p \geq 1$ and $C \geq 1/d$, $DSW_p(\cdot, \cdot; C)$, $SW_p$, $maxSW_p$, and $W_p$ are topologically equivalent, namely, the convergence of probability measures under $DSW_p(\cdot, \cdot; C)$ implies the convergence of these measures under other metrics and vice versa.*

The proof of Theorem 2 is in Appendix A.2. As a consequence of Theorem 2, the statistical error of estimating the unknown distribution based on the empirical distribution of $n$ i.i.d data under DSW distance is $C_d \cdot n^{-1/2}$ with high probability where $C_d$ is some universal constant depending on dimension $d$ (see Appendix B.3). Therefore, as other sliced-based Wasserstein distances, the DSW distance does not suffer from the curse of dimensionality.

## 3.2 COMPUTATION OF DSW DISTANCE

Direct computation of DSW distance is challenging. Hence we consider a dual form of DSW distance and a reparametrization of $\sigma$ as follows.

**Definition 2.** *For any $p \geq 1$ and admissible $C > 0$, there exists a non-negative constant $\lambda_C$ depending on $C$ such that the dual form of DSW distance takes the following form*

$$DSW_p^*(\mu, \nu; C) = \sup_{\sigma \in \mathbb{M}} \left\{ \left( \mathbb{E}_{\theta \sim \sigma} \left[ W_p^p(\mathcal{R}I_\mu(\cdot, \theta), \mathcal{R}I_\nu(\cdot, \theta)) \right] \right)^{\frac{1}{p}} - \lambda_C \mathbb{E}_{\theta, \theta' \sim \sigma} \left[ |\theta^\top \theta'| \right] \right\} + \lambda_C C,$$

*where $\mathbb{M}$ denotes the space of all probability measures on the unit sphere $\mathbb{S}^{d-1}$.*

By the Lagrangian duality theory, $DSW_p(\mu, \nu; C) \geq DSW_p^*(\mu, \nu; C)$ for any $p \geq 1$ and admissible $C > 0$. In Definition 2, the set $\mathbb{M}_C$ disappears and $\lambda_C$ plays the tuning role for the regularized term $\mathbb{E}_{\theta, \theta' \sim \sigma}\left[ |\theta^\top \theta'| \right]$. When $\lambda_C$ is large, $\mathbb{E}_{\theta, \theta' \sim \sigma}\left[ |\theta^\top \theta'| \right]$ needs to be small, meaning that $C$ is small. When $\lambda_C$ is small, the value of $\mathbb{E}_{\theta, \theta' \sim \sigma}\left[ |\theta^\top \theta'| \right]$ becomes less important, i.e., $C$ is large.

For reparametrizing the measure $\sigma$, we use a pushforward of uniform measure on the unit sphere through some measurable function $f$. In particular, let $f$ be a Borel measurable function from $\mathbb{S}^{d-1}$ to $\mathbb{S}^{d-1}$. For any Borel set $A \subset \mathbb{S}^{d-1}$, we define $\sigma(A) = \sigma^{d-1}(f^{-1}(A))$, where $\sigma^{d-1}$ is the uniform probability measure on $\mathbb{S}^{d-1}$. Then for any Borel measurable function $g : \mathbb{S}^{d-1} \to \mathbb{R}$, we have $\int_{\theta \sim \sigma} g(\theta) d\sigma(\theta) = \int_{\theta \sim \sigma^{d-1}} (g \circ f)(\theta) d\sigma^{d-1}(\theta)$. Therefore, we obtain the equivalent dual form of DSW as follows:

$$DSW_p^*(\mu, \nu; C) = \sup_{f \in \mathcal{F}} \left\{ \left( \mathbb{E}_{\theta \sim \sigma^{d-1}} \left[ W_p^p\big( \mathcal{R}I_\mu(\cdot, f(\theta)), \mathcal{R}I_\nu(\cdot, f(\theta)) \big) \right] \right)^{1/p} \right. \tag{2}$$

$$\left. - \lambda_C \mathbb{E}_{\theta, \theta' \sim \sigma^{d-1}} \left[ \left| f(\theta)^\top f(\theta') \right| \right] \right\} + \lambda_C C := \sup_{f \in \mathcal{F}} DS(f),$$

where $\mathcal{F}$ is a class of all Borel measurable functions from $\mathbb{S}^{d-1}$ to $\mathbb{S}^{d-1}$.

**Finding the optimal $f$:** We parameterize $f$ in the dual form (2) by using a deep neural network with parameter $\phi$, defined as $f_\phi$. Then, we estimate the gradient of the objective function $DS(f_\phi)$ in equation (2) with respect to $\phi$ and use stochastic gradient ascent algorithm to update $\phi$. Since there are expectations over uniform distribution in the gradient of $DS(f_\phi)$, we use the Monte Carlo method to approximate these expectations. Note that, we can use the fixed point from the stochastic ascent algorithm to approximate the dual value of DSW in equation (2). A detailed argument for this point is in Appendix B.2. Finally, in generative model applications with DSW being the loss function, we only need to use the gradient of the function $DS(.)$ to update the parameters of interest. Therefore, we can treat $\lambda_C$ as a regularized parameter and tune it to find suitable value in these applications.

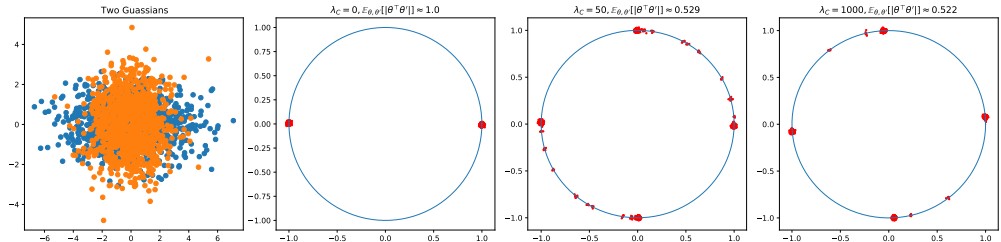

Figure 1: Empirical behavior of optimal measure $\sigma$, approximated by 1000 samples, on a circle for different values of $\lambda_C$ (the constant in the dual form of DSW in Definition 2) when $\mu$ and $\nu$ are bivariate Gaussian distributions sharing the same eigenvectors. When $\lambda_C = 0$, $C = 1$. When $\lambda_C$ increases, $C$ becomes small.

**Illustration of the roles of $\lambda_C$ and $C$:** To illustrate the roles of $\lambda_C$ and $C$ in finding optimal distribution $\sigma$, we conduct a simple experiment on two Gaussian distributions with zero means and covariance matrices given by $\begin{pmatrix} 2 & 0 \\ 0 & 2 \end{pmatrix}$ and $\begin{pmatrix} 5 & 0 \\ 0 & 1 \end{pmatrix}$. The experiment optimizes the empirical form of Definition 2 with different choices of $\lambda_C$. The results are shown in Figure 1 with the reported value of $\lambda_C$ and $\mathbb{E}_{\theta,\theta' \sim \sigma}\left[ |\theta^\top \theta'| \right]$. For $\lambda_C = 0$, the obtained distribution concentrates only on one direction. When $\lambda_C = 50$, optimal $\sigma$ distributes more weights to other directions on the circle. When $\lambda_C = 1000$, optimal $\sigma$ is close to the discrete distribution concentrated on two eigenvectors of the covariance matrices, which are the main directions differentiating the two Gaussian distributions.

**Extension of DSW and comparison of DSW to Max-GSW-NN:** Similar to SW, we extend DSW to distributional generalized sliced Wasserstein (DGSW) by using the non-linear projecting operator via GRT. The definition of the DGSW and its properties are in Appendix C. Finally, in Appendix E.1, we show the distinction of the DSW to Max-GSW-NN (Kolouri et al., 2019) when the neural network defining function in Max-GSW-NN is $g(x, \theta) = \langle x, f(\theta) \rangle$ where $f : \mathbb{S}^{d-1} \to \mathbb{S}^{d-1}$.

## 4 EXPERIMENTS

In this section, we conduct extensive experiments comparing the performance in both generative quality and computational speed of the proposed DSW distance with other sliced-based distances, namely the SW, Max-SW, Max-GSW-NN (Kolouri et al., 2019) and projected robust subspace Wasserstein (PRW) (Paty & Cuturi, 2019; Lin et al., 2020) using the minimum expected distance estimator (MEDE) (Bernton et al., 2019) on MNIST (LeCun et al., 1998), CIFAR10 (Krizhevsky, 2009), CelebA (Liu et al., 2015) and LSUN (Yu et al., 2015) datasets. The details of the MEDE framework are described in Appendix D. We would like to note that the wall-clock timing of different methods may be subject to the differences in the hyperparameter settings and software implementations of different methods. On MNIST dataset, we train generative models with different distances and then evaluate their performances by comparing Wasserstein-2 distances between 10000 random generated images and all images from the MNIST test set. Due to the very large size of other datasets, e.g., 3 million images in LSUN, it is expensive to compute empirical Wasserstein-2 distance as its complexity is of order $\mathcal{O}(k^2 \log k)$ where $k$ is the number of support points. Therefore, after we train generative models, we use FID score (Heusel et al., 2017) to evaluate the generative quality of these generators. The FID score is calculated from 10000 random generated images and all training samples using precomputed statistics in (Heusel et al., 2017). Finally, for $\lambda_C$ in DSW (see Definition 2), it is chosen in the set $\{1, 10, 100, 1000\}$ such that its Wasserstein-2 (FID score) (between 10000 random generated images and all images from corresponding validation set) is the lowest among the four values. Detailed experiment settings are in Appendix G. Finally, we also apply the DSW into color transfer task (Rabin et al., 2010; 2014; Bonneel et al., 2015; Perrot et al., 2016) in Appendix F, where we find that DSW also performs better than SW and Max-SW in this task.

### 4.1 RESULTS ON MNIST

**Generative quality and computational speed:** We report the performance of the learned generative models for MNIST in Figure 2(a). To plot this figure, we vary the number of projections $N \in$

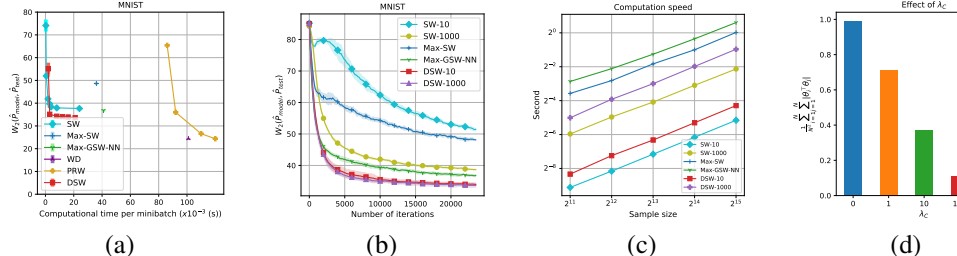

(a)  (b)  (c)  (d)

Figure 2: (a) Comparison between DSW, SW, Max-SW, Max-GSW-NN, PRW and WD based on execution time and performance. Here, each dot of SW and DSW corresponds to the number of projections chosen in $\{1, 10, 10^2, 5 \times 10^2, 10^3, 5 \times 10^3, 10^4\}$. Each dot of PRW corresponds to the dimension of the subspace chosen in $\{2, 5, 10, 50\}$; (b) Comparison between SW, DSW, Max-SW and Max-GSW-NN based on Wasserstein-2 distance between distributions of learned model and test set over iterations; (c) Computation speed of distances based on the number of minibatch's samples (log-log scale); (d) Effect of $\lambda_C$ on the mean of absolute values of pairwise cosine similarity between 10 random directions from the found distribution $\sigma$ of DSW.

$\{1, 10, 10^2, 5 \times 10^2, 10^3, 5 \times 10^3, 10^4\}$ for the SW, and $N \in \{1, 10, 10^2, 5 \times 10^2, 10^3, 5 \times 10^3\}$ for the DSW. Then we measure the computational time per minibatch and the Wasserstein-2 score of the learned generators for each $N$. We plot the Wasserstein-2 score and computational time of Max-SW and Max-GSW-NN in their standard settings (Kolouri et al., 2019). Except for the regime with very fast but low-quality learned models, DSW is better than all the existing slice-based baselines in terms of both model quality and computational speed. Moreover, DSW can learn good models with very few projections, e.g., DSW-10 achieves better model quality than Max-GSW-NN and Max-SW and is one order-of-magnitude faster than these sliced distances. Finally, with a similar computational time, a learned generator by DSW has the Wasserstein-2 score that is roughly $10\%$ lower than the one got from SW. For the qualitative comparison between these distances, we show random generated images from their generative models in Figure 7 in Appendix E.1. We observe that generated images from DSW are sharper and easier to classify into numbers than those from other baseline distances.

**Comparison with projected robust subspace Wasserstein (PRW) and Wasserstein distance:** In Figure 2(a), we plot the Wasserstein-2 score and computational time of Wasserstein distance (WD) and PRW, where the subspace dimension of PRW varies in the range $\{2, 5, 10, 50\}$. PRW is able to improve upon the model quality of slice-based methods including DSW, however at the cost of being an order of magnitude slower than DSW with 10 projections (DSW-10). We observe that DSW-10 obtains a better Wasserstein-2 score than PRW with 5-dimensional subspace, while its corresponding computational time is 30 times faster than that of PRW-5. Using 50 dimension, PRW's Wasserstein-2 score improves about $29\%$ to that of DSW-10 but the computational cost is also around 40 times slower. The model trained by WD gives good Wasserstein-2 score; however, it is computational expensive (about 40 times slower than DSW-10). The main computational advantage of DSW comes from the exact calculation of Wasserstein distance in one-dimension. The visual comparison between PRW, WD and DSW based on their generated images is in Figure 12 in Appendix E.2.

**Convergence behavior:** Figure 2(b) shows that DSW learns better models at a faster speed of convergence than other baseline distances with a very small number of projections, e.g., DSW-10 is the second lowest curve compared to curves from other sliced-based distances.

**Scalability over sample size of minibatch:** Results in Figure 2(c) show that DSW has a computational complexity of the order $\mathcal{O}(k \log k)$, which is similar to those of other sliced-based distances, where $k$ is the number of samples per batch.

**Effect of the regularization parameter** $\lambda_C$**:** For each value of $\lambda_C \in \{1, 10, 100, 1000\}$, we find the optimal distribution $\sigma$ of DSW with $N = 10$ projections, and then calculate $A_N = \frac{1}{N^2} \sum_{i,j=1}^{N} |\theta_i^\top \theta_j|$, an approximation of the regularized term $\mathbb{E}_{\theta, \theta' \sim \sigma} \left[ |\theta^\top \theta'| \right]$ in the dual form of DSW in equation (2), where $\{\theta_i\}_{i=1}^{N} \sim \sigma$. The results are shown in Figure 2(d). We observe that when $\lambda_C$ increases, $A_N$ goes down. When $\lambda_C = 0$, i.e., no regularization, $A_N$ gets close to 1, meaning that all projected directions collapse to one direction. When $\lambda_C = 1000$, $A_N$ is close to 0.1, suggesting that all projected directions are nearly orthogonal.

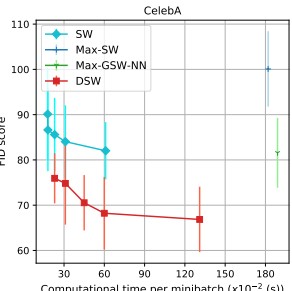 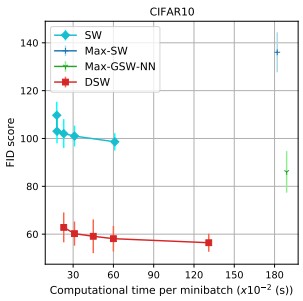 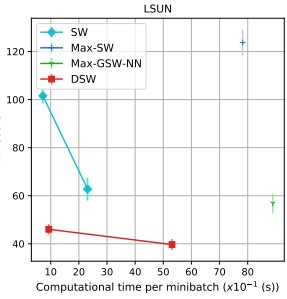

Figure 3: Comparison between DSW, SW, Max-SW and Max-GSW-NN in terms of execution time and performance. Here, each dot of SW and DSW corresponds to the number of projections chosen in $\{10^2, 5 \times 10^2, 10^3, 5 \times 10^3, 10^4\}$. We set the minibatch size be 512 on CelebA and CIFAR, and be 4096 on LSUN.

**Additional experiments:** We also investigate how the number of gradient-steps used for updating distribution of directions $\sigma$, and how the size of minibatches affects the quality of DSW (see Appendix E.1). The results show that an increasing number of gradient steps to update $\sigma$ leads to better performance of DSW but also slows down the computation speed. Furthermore, we carry out experiments with DGSW, an extension of DSW to non-linear projections, and test the new proposed distances in training encoder-generator models on MNIST using joint contrastive inference (JCI) in Appendices E.1 and E.3. The description of these models is in Appendix D.

## 4.2    RESULTS ON LARGE-SCALE DATASETS

Next, we conduct large-scale experiments on a range of more realistic image datasets. We train generative models using CIFAR10, CelebA, and LSUN datasets (all these datasets are rescaled to 64x64 resolution). When working with high dimensional distributions, Deshpande et al. (2018) proposed a trick to improve the quality of the generator by learning a feature function which maps data to a new feature space that is more manageable in size. When the feature function is fixed, the generator is trained to match the distribution of features. When the generator is fixed, the feature function tries to tease apart the data empirical features from the generated feature distribution. For the experiments in this section, we use the same technique with DSW and all other baseline distances.

We compare DSW with SW, Max-SW, and Max-GSW-NN in both generative quality (FID score) and computational time in Figure 3. We could not compare DSW with PRW on the large-scale datasets since PRW is computationally expensive to train to obtain good generated images. On CelebA and CIFAR10, we let $N$, the number of projections of both DSW and SW, vary in the set $\{10^2, 5 \times 10^2, 10^3, 5 \times 10^3, 10^4\}$. For LSUN, since it takes considerably longer time to train each model, we only vary $N$ in the set $\{10^2, 10^4\}$. On all these large datasets, DSW outperforms all the other baselines in both FID score of the learned model and computational efficiency. The gap of FID scores between DSW and other methods is especially large on CIFAR10 and LSUN. For example, on CIFAR10, with the same computational time, FID scores of DSW are always lower than those of SW about 20 units. On LSUN, with 100 projections, DSW can achieve an FID score of 46 while SW with 10000 projections still has a worse FID score of over 60. It is interesting to note that on these high-dimensional datasets, Max-SW performs rather poorly: it obtains the highest FID scores among all distances while requires heavy computation. Max-GSW-NN has better FID scores than (Max)-SW; however, it is still worse than DSW and while being slower. This is consistent with the intuition that as the number of dimension of the data grows, the use of a single important slice in Max-SW becomes a less efficient approximation. DSW, on the other hand, is able to make use of more important slices, and at the same time avoids SW's inefficiency of uniform slice-sampling.

Generated images from CelebA, CIFAR10 and LSUN are deferred to Appendix E.1. Comparing to other sliced-based Wasserstein distances, generated samples obtained from the DSW's generative model are also more visually realistic. Further experiments to compare DGSW with GSW, Max-GSW, and Max-GSW-NN are also given in the Appendix E.1. Based on these experiments, we can conclude that the distributional approach also improves the generative quality of non-linear slicing distances.

## 5   CONCLUSION

In this paper, we have presented the novel distributional sliced-Wasserstein (DSW) distances between two probability measures. Our main idea is to search for the best distribution of important directions while regularizing towards orthogonal directions. We prove that they are well-defined metrics and provide their theoretical and computational properties. We compare our proposed distances to other sliced-based distances in a variety of generative modeling tasks, including estimating generative models and jointly estimating both generators and inference models. Extensive experiments demonstrate that our new distances yield significantly better models and convergence behaviors during training than the previous sliced-based distances. One important future direction is to investigate theoretically the optimal choice of the regularization parameter $\lambda_C$ such that the DSW distance can capture all the important directions that can distinguish two target probability measures well.

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
