# OpenReview forum: "Distributional Sliced-Wasserstein and Applications to Generative Modeling"
_ICLR.cc/2021/Conference — ICLR 2021 Spotlight_

### Official Review · AnonReviewer4 · 2020-10-27
**Interesting novel variant of Sliced Wasserstein, but somehow incremental**

**Rating:** 7
**Confidence:** 5

**Review:**


The paper presents a novel variant of the Sliced Wasserstein (SW) distance. Wasserstein distances have been used recently in lot of machine learning problems. One of the major problem is that, in its primal form, it is computationally expensive. In order to alleviate this problem, a class of methods, called sliced, leverage on the fact that Wasserstein has a closed form expression in 1D (which amount to sort the samples). It replaces the original Wasserstein distance by an expectation of 1D sub-problems over directions drawn uniformly on the unit hypersphere (akin to a Radon transform). Observing that not all the directions are meaningful, the authors propose a variant of SW where the expectation over all the directions is replaced by an expectation over a distribution of directions. The ‘extent’ of this distribution is controlled by an extra parameter. Interestingly, the authors show that this formulation is computationally tractable if one parametrizes this distribution by a measurable function, expressed as a neural network. This result is obtained by deriving the Lagrangian dual of the original problem. Comparisons with previous works are then given in two GAN scenarii: one on MNIST to explore the importance of the different parameters, and another on larger and more complicated datasets, where the FID score is exposed.

The paper is well written, very clear and easy to follow. Related works are correctly cited. There is one missing work that should be cited though:

Meng, Cheng, et al. "Large-scale optimal transport map estimation using projection pursuit." Advances in Neural Information Processing Systems. 2019.

Where authors choose the most relevant directions in a way similar to projection pursuit. I really believe this paper could also lead to interesting comparisons.

All in all, the presented method is a variant of SW that builds on several previous works exploring a similar idea (how to better sample the directions), such as max-SW or subspace robust Wasserstein distances. As such, it can be considered relatively incremental, but this should not totally prevent publications if the computational benefits/performances are very good. Regarding this point, I have some questions about the experimental section:
 - it seems that Figures 2.a (W_2 score) and 3.a (FID) are obtained at convergence. From appendix F, it seems that the number of training epochs is fixed for all methods. I wonder if this setting is fair for comparing max-SW, as far as only one direction is contained (the max direction), the gradient might gather less information. I suspect that more iterations might be needed, why not going until the full convergence  of the model ?
 - when comparing SW and DSW, are the directions drawn randomly for every batches ? (I guess this the common practice when using SW). As such, I do not understand what is the meaning of fixing the number of directions in advance.
 - finally it seems that DSW is computed on mini-batches of samples. While I acknowledge there is a common practice to do so, I think computing a 1D Wasserstein on a mini batch is not the same as computing the 1D W on the full dataset. As such, this mini batch version of SW is not the same as computing the true SW. In the end, i) the size of the mini-batch might have an impact on the estimation quality, that should be discussed ii) if computing 1D W on mini batches, why not computing and comparing with the mini batch version of the original version of W ? This has been done in several papers and has shown to give good results (See the recent study on this theme
Fatras, Kilian, et al. "Learning with minibatch Wasserstein: asymptotic and gradient properties." the 23nd International Conference on Artificial Intelligence and Statistics. Vol. 108. 2020.
) , plus it does not have the limitation/artefacts of the sliced Wasserstein.

Finally, I would have liked to see other types of experiments than a GAN to really assess the power of this new methodology (SW has been used for classification, domain adaptation, computing gradient flows, etc.). Why limiting the applications to generative modeling ? If generative modeling is the target, then I would expect to see comparisons with other types of metrics than SW (wether it be dual version of W_1 as in WGAN, MMD, or any other divergence from the model zoo).

For all these reasons my recommendation will only be ‘above acceptance threshold’.

## after author response
I thank the authors for providing sensible answers to my questions, and analysis of their method compared to the minibatch version. I am still not sure about the fact that the rate of convergence of  SW is indeed indepedent of the dimension, as the dimension is indeed hiddent in the scalar product. Nevertheless, I believe this question is out of the scope of the paper, and I changed my final rating accordingly to the new version of the manuscript.

---

> ### Author Response · Authors · 2020-11-16
> **Author response - Part 1**
>
> Thank you for your time. We have revised our papers based on your comments. The changes are marked in blue color.
>
> Question 1: “Comparing with Meng, Cheng, et al. "Large-scale optimal transport map estimation using projection pursuit." Advances in Neural Information Processing Systems. 2019.”
>
> Answer:  Thank you for your reference. Based on your suggestion, we have implemented two variants of SW, which are based on projection pursuit methods (e.g., directional regression (DR) and sliced average variance estimator (SAVE)) to find the most “informative” projecting directions. We name these variants as drSW and saveSW. In deep generative modeling tasks, with the minibatch setting, DR and SAVE cannot be applied because they require to inverse the empirical covariance matrix, which is normally low rank in the high dimensional setting. Therefore, we only apply drSW and saveSW to the color transfer tasks. The experiment results are in Figures 16 and 17 in Appendix F. Given these results, we observe that both drSW and saveSW perform worse than max-SW, SW, and DSW. Furthermore, DSW produces the most lively and realistic transferred images among these baselines.
>
> Question 2: “It seems that Figures 2.a (W_2 score) and 3.a (FID) are obtained at convergence. From appendix F, it seems that the number of training epochs is fixed for all methods. I wonder if this setting is fair for comparing max-SW, as far as only one direction is contained (the max direction), the gradient might gather less information. I suspect that more iterations might be needed, why not going until the full convergence of the model?”
>
> Answer:  Thank you for your comment. In the revised version, we have increased the number of epochs for Max-SW to 800 and run the generative model based on Max-SW on MNIST. The experiment results are in Figure 4(c) in Appendix E.1. Based on the results, Max-SW’s Wasserstein-2 score increases considerably with more iterations; however, Max-SW’s result is still worse than that of DSW with a much smaller number of iterations. This result suggests that Max-SW might need several more iterations than DSW to obtain a comparable quality generative model.
>
> Question 3: “When comparing SW and DSW, are the directions drawn randomly for every batches ? (I guess this the common practice when using SW). As such, I do not understand what is the meaning of fixing the number of directions in advance.”
>
> Answer: Thank you for your question. We would like to clarify here that the directions are drawn randomly for every batch. By fixing the number of projections, it means that we do not change the number of drawn directions in each minibatch.
>
>
> Question 4:  “The size of the mini-batch might have an impact on the estimation quality, that should be discussed.”
>
> Answer: Thank you for your suggestion. We have already done this in Figure 5(b) in the Appendix. We observed that having a bigger size of minibatch in the case of DSW can help the model distribution to converge faster to the data distribution in the sense of Wasserstein-2 distance. However, the differences are not significant when changing the minibatch size in the set  {128,256,512,1024,2048}.

---

> > ### Comment · ~Cheng_Meng1 · 2021-02-24
> > **Question about the color transfer results of saveSW and drSW**
> >
> > This is an interesting paper, and I enjoy reading it.
> >
> > I am one of the authors of the paper "Large-scale optimal transport map estimation using projection pursuit," and we appreciate you citing our work. We read the appendix and are shocked to find that our proposed methods, i.e., "saveSW" and "drSW", work badly in the color transfer application, shown in Fig. 16~17. We are suspicious that our methods have not been implemented correctly.
> >
> > We replicate the color transfer applications using our implementation and observe that "saveSW" and "drSW" work reasonably well. The code and the results can be found in my GitHub (https://github.com/ChengzijunAixiaoli/PPMM/blob/master/color%20transfer.ipynb). We appreciate it if you could update the result in the appendix and make an objective evaluation of our method.
> >
> > Please feel free to contact me if you have any questions. Thanks.

---

> > > ### Comment · ~Nhat_Ho1 · 2021-02-24
> > > **Re: color transfer results**
> > >
> > > HI Cheng,
> > >
> > > We thank you for your comment. For the implementation of drSW and saveSW, we also used your repo in the given link. We have spent time figuring out the reason for this difference in visual results by looking at your color transfer code and ours, then we realized the following reason. In our experiment, we use this repo to do the color transfer (https://github.com/BorisMuzellec/SubspaceOT/blob/master/Color%20Transfer.ipynb). In this repo, after finding the transportation map between the compressed source image and the compressed target image, the transferred image is created by moving the color from the source to the target. This process is done only one time and it is not iterative. This equals to set the number of iterations to 1 in your color transfer implementation.
> > >
> > > We have conducted experiments again using your framework (we put the modified code in the supplementary) and updated the results in Figure 17 and Figure 18 in the appendix of our paper.  We found that DSW is still better than SW and Max-SW. However, it is not clear which is the best method among DSW, saveSW, and drSW, since DSW gives images that are similar to images from EMD (Earth Mover Distance), while saveSW and drSW provide smoother images.
> > >
> > > Finally, we thank you again for your comment. Please let us know if you still have any other concerns. Thank you very much.
> > >
> > > Best,
> > >
> > > Nhat Ho

---

> > > > ### Comment · ~Cheng_Meng1 · 2021-03-22
> > > > **Re:Re: color transfer results**
> > > >
> > > > Hi Nhat,
> > > >
> > > > Thanks for the prompt response. The latest repo addressed the issue and the results make a lot of sense now.
> > > > We appreciate your hard work!
> > > >
> > > > Best,
> > > >
> > > > Cheng Meng

---

> ### Author Response · Authors · 2020-11-16
> **Author response - Part 2**
>
> Question 5: “If computing 1D W on mini batches, why not computing and comparing with the mini batch version of the original version of W ? This has been done in several papers and has shown to give good results (See the recent study on this theme Fatras, Kilian, et al. "Learning with minibatch Wasserstein: asymptotic and gradient properties." the 23nd International Conference on Artificial Intelligence and Statistics. Vol. 108. 2020. ) , plus it does not have the limitation/artefacts of the sliced Wasserstein.”
>
> Answer: Thank you for your insightful comment. We have included a comparison between DSW and Wasserstein distance with different minibatch's size in Table 5 in Appendix E.2. On MNIST, we observe that with minibatch size of 512, the Wasserstein distance is slower than the DSW (40 times slower) though its Wasserstein-2 score is better than that of DSW (from 34.4 in DSW to 24.4 in Wasserstein distance). When the minibatch size is 1024, the Wasserstein is about 75 times slower than DSW while its Wasserstein-2 score only improves slightly comparing to its value when the minibatch size is 512 (cf. Table 5 in Appendix E.2 for more details). We would expect that when the minibatch size is larger, the computational cost of Wasserstein distance will increase considerably while that of distributional sliced Wasserstein distance is still quite cheap.
>
> Finally, though sliced Wasserstein distance has some limitations, we would like to take this opportunity to explain two main benefits of using sliced-based Wasserstein distances, including DSW, over the Wasserstein distance:
>
> The first benefit is from the statistical gain. In particular, the sliced-based Wasserstein distances do not suffer from the curse of dimensionality as that of the Wasserstein distance. It means that if our data lie in $d$ dimensions, the rate of convergence of the model under Wasserstein distance is proportional to $n^{-1/d}$ (cf. the reference [1]) where $n$ is the number of data, which can be costly when the dimension $d$ is sufficiently large. However, for the sliced-based Wasserstein distances, including the DSW, they do not suffer from the curse of dimensionality, namely, the model converges at the rate $n^{-1/ 2}$, which is free from the dimension even when it is very large. It means that we require fewer number of samples for the sliced-based Wasserstein distances compared to the Wasserstein distance.
>
> The second benefit is from the computational gain, which is also one of the main motivations of using sliced-based Wasserstein distance in the training. In particular, solving the Wasserstein distance is famously known for its expensive computational complexity as it is equivalent to solving a linear programming problem. Such problem has the best practical complexity of the order $N^3$ (cf. the reference [2]), where $N$ stands for the number of supports of each probability measure. To account for this expensive computation of solving directly the Wasserstein distance, an entropic regularization version of Wasserstein distance, named entropic regularized Wasserstein, had been proposed (cf. the reference [3]). The computational complexity of entropic regularized Wasserstein for approximating the original Wasserstein distance is at the order $C(d) N^2$ (up to some polynomial order of approximation error) where $C(d)$ is some constant depending on $d$. Due to its computational benefits, several recent papers used entropic regularized Wasserstein and its divergence, named Sinkhorn divergence, to train deep learning that involves Wasserstein distance as the loss function. However, $C(d) N^2$ can be still quite expensive since $d$ is usually large and $N$ is not small in large-scale datasets and since we need to compute the entropic regularized Wasserstein repeatedly in the training. For this reason, another line of works is to use sliced-based Wasserstein distances, which have a computational complexity of the order $N \log N$, to deal with this problem of training of Wasserstein distances and their entropic versions.
>
> In summary, sliced-based Wasserstein distances have both statistical and computational benefits. However, as you mentioned, we would have the loss of information of the original measures from projecting them to one dimension space, which can be thought as a trade-off between computational time and quality of the results. The introduction of DSW is to balance this trade-off by trying to keep as much important information as we could through finding several important directions that can distinguish the probability measures while having computational complexity that is of the order $N \log N$.
>
> [1] Fournier, N. & Guillin, A. On the rate of convergence in Wasserstein distance of the empirical measure. Probability Theory and Related Fields 162, 707–738, 2015.
>
> [2] O. Pele & M. Werman. Fast and Robust Earth Mover’s Distances. ICCV, 2009.
>
> [3] M. Cuturi. Sinkhorn Distances: Lightspeed Computation of Optimal Transport. NIPS, 2013.

---

> > ### Comment · AnonReviewer1 · 2021-03-28
> > **with regard to proof of sample complexity**
> >
> > In the proof of Theorem 4 in the supplement, there is an equality sign between the expression involving inverse CDFs and the CDFs, but this equality only holds for $p=1$. There should be a preceding inequality based on the fact that the $L_1$ norm is greater than or equal to $L_p$ norms for $p\ge 1$.

---

> ### Author Response · Authors · 2020-11-16
> **Author response - Part 3**
>
> Question 6:  “Finally, I would have liked to see other types of experiments than a GAN to really assess the power of this new methodology (SW has been used for classification, domain adaptation, computing gradient flows, etc.). Why limiting the applications to generative modeling ? If generative modeling is the target, then I would expect to see comparisons with other types of metrics than SW (wether it be dual version of W_1 as in WGAN, MMD, or any other divergence from the model zoo).”
>
> Answer: Thank you for your suggestion. In generative modeling, (Deshpande et al 2018) [4] had already compared SW with the dual version of W_1 in WGAN. They showed that SW is robust to the generator's architecture while WGAN failed in many types of architecture. Moreover, we agree that the DSW distance can be used in other applications. In the revised version, we have applied the DSW to the color transfer task and compared DSW to SW and Max-SW (see Figure 16 in Appendix F). In the experiments, we find that SW and DSW create smoother images compared to Max-SW, which creates some noise parts in its transferred images. Furthermore, DSW produces more lively and realistic images than SW, especially when the number of projections is small (e.g., 10 projections).
>
> [4] I. Deshpande & Z. Zhang & A. Schwing. Generative Modeling using the Sliced Wasserstein Distance. CVPR, 2018.

---

### Official Review · AnonReviewer1 · 2020-10-28
**Slicing from a tuned distribution of slices is better than one best slice or uniform slicing**

**Rating:** 7
**Confidence:** 4

**Review:**

Summary:
The paper describes a family of sliced Wasserstein divergences that maximize the distribution over slices subject to constraints on the concentration of slices. Extremes of the family are the sliced Wasserstein and max-sliced Wasserstein distance. In between these the divergence is sensitive to informative discrepancies in multiple subspaces, while still leveraging the relatively fast computation the Wasserstein distance in one dimension for empirical distributions. A dual formulation provides a variational approximation using a (possibly deep) neural network to instantiate the slicing distribution through a pushforward approach. Basic theory prove the divergence is a distance metric between measures. Extensive experiments show improvement over sliced and max-sliced Wasserstein distances and related projection based approaches.

Strengths:
Overall the paper is a clear and original contribution to the field of sliced Wasserstein distances. The paper's appendix shows multiple applications of the methodology where it outperforms existing divergences.

Weaknesses:
The specific parameterization of the mapping $f$ is not clear (it is not given in the main body and it is poorly described in Appendix F as a "single multi layer perceptron (MLP) layer with normalized output"). In the main body it is called a deep neural network.

Also I am a bit wary of interpreting the wall-clock timing differences due to differences in software implementation and hyper-parameter choices for optimization and number of gradient steps (Figure 2(a,c) and Figure 3). Learning curves are more fair, but even these use the best value of  $\lambda_C$ versus default settings for other methods. As the paper recognizes the max sliced generalized Wasserstein with a deep neural network is the closest competitor and also requires solving a max-min saddle point optimization. It is shown in the appendix as Figure 6(a–e) (green trace). Even in this case the differences in learning curve and timing could possibly be due to software implementation and hyper-parameter selection in the optimization.

Overall this is a clear accept as it a meaningful improvement to sliced Wasserstein approaches. There a few points that need to be clarified, and I would argue for less emphasis on the wall-clock timing of methods.

How many parameters are used in the neural network defining $f_\phi$? Would an empirical distribution optimized over a corresponding number of fixed slices (originally randomly drawn) perform as well?

Minor comments (main body):
P.1 use Author (year) instead of Author (Author year).

P.1  The case of empirical sample has not been described so what is meant by the term "atoms" to describe $k$ is not apparent to the reader.  Then on page 6 "$k$ is the number of support points". The terminology could be more consistent.

P.2  Double parentheses "))"  after Deshpande et al., 2019.

References. Capitalization of conference names and book titles is not consistent. Sinkhorn is a proper noun and should be capitalized.

Minor comments (appendix):

The appendix doesn't appear to be carefully proofread.

Sometimes the notation uses $.$ instead of $\cdot$ for the arguments of multivariate functions, this is harder to read as in  $\mathrm{DSW}_p(.,.;C)$ versus $\mathrm{DSW}_p(\cdot,\cdot;C)$

There are a number of missing definite articles for example "that length of side of right triangle" -> "that the length of the side of the right triangle".

Throughout, "close-form" -> "closed-form".

P.18 "subspcaae" and full stop "." beginning new line.

P.19 "as the al time increases considerably"

P.19 " DSW and DGSW are slower than Max-SW, Max-GSW (50 gradient updates to find the max direction), and Max-GSW-NN (50 update times for the defining neural net function)."  Is there any meaning to the subtle variation between "gradient updates" and "update times".

P.23 "single multi layer perceptron (MLP) layer" ? This is not clear.

P.23  "as the f function in the dual empirical forms of DSW and DGSW for the dual empirical forms of these distances)" Redundant wording and extra parentheses.

---

> ### Author Response · Authors · 2020-11-16
> **Author response**
>
> Thank you for your time. We have revised our papers based on your comments. The changes are marked in blue color.
>
> Question 1: “The specific parameterization of the mapping f is not clear (it is not given in the main body and it is poorly described in Appendix F as a "single multilayer perceptron (MLP) layer with normalized output"). In the main body, it is called a deep neural network.”
>
> Answer: Thank you for your comment. We apologize for this confusing statement about the mapping $f$. In fact, we want to indicate that it is quite flexible to apply various architectures to parametrize the function $f$. In practice, using a powerful neural net may require harder computation while we prefer the fast computational speed of the distance. Therefore, for the sake of computation, we parametrize $f$ as a MLP network with just one layer with $d^2$ parameters where d is the dimension of comparing distributions (with the input is normalized to provide vectors from the unit sphere). We believe that this network is expressive enough in the general case of comparing arbitrary distributions. Finally, we would like to remark that if we have more structural information about two target probability measures, special architectures of deep learning might be helpful by introducing some inductive biases.
>
> Question 2: “Wall-clock timing differences due to differences in software implementation...I would argue for less emphasis on the wall-clock timing of methods”.
>
> Answer: Thank you for your comment. We agree that the hyperparameter settings and implementations may affect the performance of the reported methods. In the revised version, at the beginning of Section 4, we have included an additional remark that the wall-clock timing differences of different methods may be subject to the differences in the hyperparameter settings and software implementations of different methods.
>
> Question 3: “Would an empirical distribution optimized over a corresponding number of fixed slices (originally randomly drawn) perform as well?”
>
> Answer: We assume that you mean that the family distribution over slices could be restricted to discrete measures that have n support points with uniform weights. Then, there is no need for the mapping $f$ anymore, i.e., we can optimize directly the n points with the regularization. In Appendix E.1 of the revised version, we have added the experiments to compare this version of DSW, which we refer to as discrete DSW (dDSW), with the general form of DSW in generative modeling tasks on MNIST. The results are reported in Figure 4(d). We observe that dDSW performs better than sliced Wasserstein (SW) and max-sliced Wasserstein (Max-SW), namely, dDSW (n=10) converges faster than SW (n=10) and Max-SW in the sense of Wasserstein-2. However, both dDSW with n=10 and n = 1000 are worse than DSW with n = 10 (and thus with n = 1000). It suggests that DSW is better than dDSW.
>
> Question 4: “Is there any meaning to the subtle variation between "gradient updates" and "update times”?”
>
> Answer: Thank you for your question. There is no variation between gradient updates and update times. In the revised version, we have changed “update times” to “gradient updates” for more consistency in terminology.
>
> Question 5: “Typos, inconsistent terminologies”.
> Thank you for your comment. We have fixed all the typos and inconsistent terminologies in the revised version.

---

### Official Review · AnonReviewer2 · 2020-10-29
**Distributional Sliced-Wasserstein and Applications to Generative Modeling**

**Rating:** 9
**Confidence:** 5

**Review:**

This is a well written paper with some interesting results.  This paper is to propose a distributional sliced-Wasserstein distance
to address the limitations of standard SW and Max-SW. The proposed method finds an optimal distribution over projections that can balance between exploring distinctive projecting directions and the informativeness of projections. Some theoretical results are presented in both the main paper and its supplementary document. This reviewer personally enjoys reading this paper.
Here are a few additional comments.

1. How to select $\lambda_C$ in practice? The authors need to discuss it in details.

---

> ### Author Response · Authors · 2020-11-16
> **Author response**
>
> Thank you for your comment. We have mentioned how to select $\lambda_{C}$ at the beginning of Section 4 on Page 6. In particular, on the MNIST dataset, we split the test set of MNIST into a validation set and a test set with 20% samples and 80% samples respectively. Then, we search for $\lambda_{C} \in $ {1, 10, 100, 1000}, which gives the best Wasserstein-2 score on the validation set and then use that value of $\lambda_C$ to evaluate the score on the test set. Similarly, on CelebA, CIFAR10, and LSUN, we also split the test set with the same ratio as that in MNIST, and also search for $\lambda_C \in $ {1, 10, 100, 1000}  that gives the best FID score. In the experiments, we observed that the optimal value of $\lambda_C$ is always either 1 or 10. Therefore, we would suggest choosing $\lambda_C$ between 1 and 10 in other practical applications of DSW.
>
> Finally, given the above observation, we would like to remark that investigating theoretically the optimal choice of the regularization parameter $\lambda_{C}$ such that the DSW distance can capture all the important directions that can distinguish two target probability measures well is an important and interesting direction that we will pursue in the future.

---

### Author Response · Authors · 2020-11-24
**Look forward to your final feedback**

Dear reviewers,

Thanks for spending your time reading our paper and providing the insightful comments.

As the ending of the discussion period is approaching, we look forward to hearing your feedback regarding our rebuttals and the revision of our paper. We are happy to discuss if you still have any other concerns.

Best,

Authors

---

### Comment · ~Ammar_Fayad1 · 2022-11-28
**Question about the Constraints**

The paper is quite good, but I have a question about the constraint $\mathbb{E}_{\sigma}[|\theta^\top \theta']\leq C$. It is well-known that in high dimensions, any two random vectors are almost orthogonal, which means your expectation is almost always close to zero, so why does it make sense to enforce such constraint in high-dimensional settings? Thank you!

---

> ### Author Response · Authors · 2022-11-28
> **Thank you for your question.**
>
> Dear Ammar,
>
> It is true that uniformly random vectors are almost orthogonal. However, in the paper, we search for the distribution that can maximize the expected projected distance. Without the constraint, the measure will collapse to a Dirac Delta probability measure at the "max" vector. That is the reason that we use the orthogonality constraint. In fact, other constraints that avoid collapsing can also be used.
>
> Best regards,

---

### Decision · Program_Chairs · 2021-01-07
**Final Decision**

**Decision:**

Accept (Spotlight)

**Comment:**

In this paper, the authors propose a new max-sliced Wasserstein distance. Specifically, the proposed method is a multiple sliced variants of the existing max-sliced Wasserstein distance. Compared to the subspace Robust Wasserstein distance, the proposed method can be efficiently computed.

Overall, the proposed method is a good extension of the max-sliced Wasserstein and can be used in various applications. All authors agree to accept the paper, so, I also vote for acceptance.